# ViT-UWA: Vision Transformer Underwater-Adapter for Dense Predictions Beneath the Water Surface

## Abstract

Vision Transformer (ViT) and its variants have witnessed a significant success in computer vision. However, they do not perform well in underwater dense prediction tasks due to challenges like complex underwater environments, quality degradation, and light scattering in underwater images. To solve this problem, we propose the Vision Transformer Underwater-Adapter (ViT-UWA), the first detail-focused and adapted ViT backbone for underwater dense prediction tasks, without requiring task-specific pretraining. In ViT-UWA, we first introduce High-frequency Components Prior (HFCP) to add high-frequency information of underwater images to the plain ViT, which can help recover and capture lost high-frequency information of underwater images. Then, we propose an Detail Aware Module (DAM) to obtain a detail-focused multi-scale convolutional feature pyramid, which can be used in kinds of dense prediction tasks. Through the ViT-CNN Interaction Module (VCIM), we achieve bidirectional feature fusion between ViT and CNN. We evaluate ViT-UWA on multiple underwater dense prediction tasks, including semantic segmentation, instance segmentation, and object detection. Notably, with only ImageNet-22K pretraining, our ViT-UWA-B yields state-of-the-art **46.4** box AP and **44.2** mask AP on USIS10K dataset. We hope ViT-UWA could provide a new backbone for future research on underwater dense prediction tasks.

## 1 Introduction

In recent years, with the increasing demand for underwater robot target capture and people's emphasis on the utilization of marine resources, there has been a growing focus on the field of underwater vision [28]. Dense prediction tasks are a type of task in the field of computer vision that involves making predictions for each pixel or small region of an image. Dense prediction tasks typically require classification or regression of each pixel of an image and effective multi-scale feature representation for classifying or detecting objects or regions with varying sizes [31]. Dense prediction tasks like semantic segmentation, instance segmentation, and object detection have significant application value in many underwater vision scenarios, such as visually-guided underwater robot [26], mapping and monitoring of marine habitats [41], underwater target detection and segmentation [29].

Inspired by the success of transformers in Natural Language Processing (NLP), vision transformers [12] soon attracted attention and rose in many computer vision tasks such as image classification, semantic segmentation, and object detection, outperforming CNN models and reaching state-of-the-art (SOTA) performance. These models are mainly split into three branches: the plain ViT [12, 35], vision-specific variants (e.g., SegFormer [58], Swin [40], PVT [53]) and adapted ViT backbones (e.g., ViT-Adapter [8], ViT-CoMer [56]). The plain ViT optimizes the use of ViT features without changing the framework of ViT. The vision-specific variants

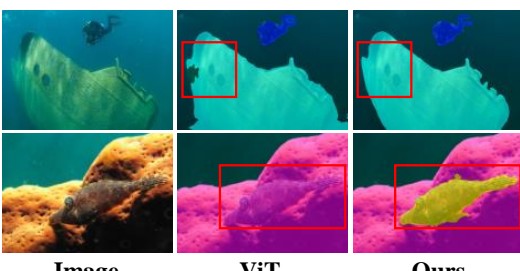

| Image | ViT | Ours |

Figure 1: **A simple comparison of ViT and ViT-UWA on the SUIM dataset.** Red boxes in the second column show several ViT's issues on underwater dense prediction.

redesign the network structure by combining the advantages of CNN and Transformer. Adapted ViT backbones only introduce CNN features by adding a parallel network, which leverages various open-source pre-trained ViT weights and addresses the lack of interaction among local ViT features and the limitation of single-scale representation. Adapted ViT backbones have made a remarkable process in dense prediction tasks.

However, due to the uneven illumination, monotonous color, and complicated underwater background of underwater images [28], underwater images usually suffer from quality degradation issues and lose a large amount of high-frequency detail information. Thus, plain ViT often encounters issues such as edge blurry in segmentation and detection, and incorrect category prediction (as shown in Figure 1) in underwater dense prediction tasks. Existing methods (e.g., Sea-Thru [1], WaterGAN [32]) address underwater degradations like color distortion based on the physics of underwater light scattering, but most of them are primarily focused on underwater image enhancement rather than underwater dense prediction. Recent studies (e.g., SUIM-Net [26], WaterMask [37], USIS-SAM [36]) usually focus on a single underwater task, which is not universal enough. Underwater dense prediction is still a challenging task for ViT and adapted ViT backbones.

To address the above issues and fill the gap where there are currently no universal methods for underwater dense prediction tasks, we propose the Vision Transformer Underwater-Adapter (ViT-UWA). It is an additional network that can adapt the plain ViT to downstream underwater dense prediction tasks without modifying ViT's primary structure. Specifically, we design three modules for ViT-UWA, including (1) an high-frequency components prior to recovering and capturing lost high-frequency information of underwater images, (2) a detail aware module to improve ViT's perception of high-frequency details, (3) a ViT-CNN interaction module to fuse features bidirectionally between ViT and CNN. As shown in Figure 2, our models continuously achieve improved performance compared to the plain ViT and recently adapted ViT backbones under the fair pre-training strategy.

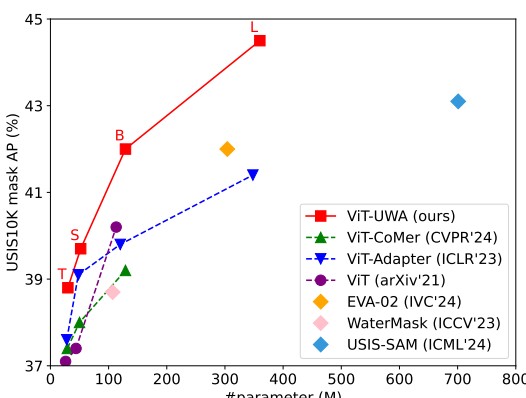

Figure 2: **Instance segmentation performance on USIS10K.** It can be seen that the proposed ViT-UWA achieves improvements to the plain ViT, adapted ViT backbones and task-specific underwater methods.

The main contributions of our work are as follows:

- We propose a novel underwater dense prediction backbone by combining the plain ViT with high-frequency components and multi-scale convolutional features. It fully leverages the prior information of underwater images and the rich semantic representation of multi-scale features, which enhances the perception of semantic boundaries in underwater images and recovers the high-frequency information in the images.

- We introduce a high-frequency components prior and design a detail aware module and a ViT-CNN interaction module. The former can help ViT recover and capture lost high-frequency information of underwater images such as edges and textures. The latter two modules can obtain detail-focused multi-scale convolutional features and perform bidirectional feature interaction between ViT and CNN, respectively.

- We evaluate the ViT-UWA on three challenging underwater dense prediction benchmarks, including SUIM [26], UIIS [37], and USIS10K [36]. Extensive experiments on public evaluation criteria demonstrate the effectiveness of the proposed ViT-UWA. For underwater image semantic segmentation, ViT-UWA-B reaches 75.3% mIoU on the SUIM dataset when using only ImageNet-1K pre-training, outperforming ViT-Adapter-B by 2.9 points and ViT-CoMer-B by 2.2 points. Moreover, for underwater object detection and instance segmentation, our ViT-UWA-B yields 30.9% box AP and 29.0% mask AP on the UIIS dataset, 46.4% box AP and 44.2% mask AP on the USIS10K dataset, which is comparable with SOTA methods.

## 2 RELATED WORK

**Vision Transformer.** In recent years, transformers have achieved significant success in multiple domains, such as natural language processing, computer vision, and audio processing. Vision Transformer (ViT) [12] first introduces the transformer to the image classification in computer vision without much modification of the original structure, achieving excellent performance. Conformer [45] first proposes a dual network to combine transformer and CNN. MAE [21] and BEiT serious [3, 44, 54] explore the potential of ViT in self-supervised learning by masked image modeling (MIM). Swin Transformer [40] designs window attention and hierarchical structure, introducing the locality of convolution operation and saving computation. However, due to quality degradation issues such as blurred edges and color cast of underwater images [18] and the weakness of single-scale representation, ViT does not perform well in underwater tasks.

**Underwater Dense Prediction.** Dense prediction tasks include semantic segmentation, instance segmentation, object detection, etc. Due to low visibility, blurred edges, low contrast, and color deviation of underwater images, underwater dense prediction is challenging for models trained on terrestrial datasets. SUIM-Net [26] introduces a fully convolutional encoder-decoder structure to balance the trade-off between performance and computational efficiency while ensuring fast end-to-end inference. UISS-Net [22] proposes an auxiliary feature extraction network and utilizes channel attention mechanism to extract multi-scale features, enhancing the segmentation ability of edge details. WaterMask [37] is the first work to explore underwater image instance segmentation, which designs a difference similarity graph attention module and a multi-level feature refinement module to reconstruct and refine the degraded image features of underwater images. USIS-SAM [36] first applies the Segment Anything Model (SAM) to the underwater salient instance segmentation task, and proposes an underwater adaptive ViT encoder and salient feature prompt generator to perform highly precise end-to-end segmentation.

**Adapted Backbones.** Adapters are originally proposed in the NLP field as an efficient method for fine-tuning large pre-trained models for each downstream task through compact and scalable models. The emergence of large-scale models has spurred the development of various adapters. Adapters [23] introduce new modules into the transformer encoder to fine-tune for specific tasks, enabling the pre-trained model to quickly adapt to downstream NLP tasks. In [47], the concept of multi-task learning is investigated, utilizing a single BERT model that was shared across several task-specific parameters. The CLIP-based adapter [60, 17] proposes transferring pre-trained knowledge to zero-shot or few-shot downstream tasks. In computer vision, VPT [27] proposes a method that freezes the pre-trained weights of ViT and updates only the parameters of the adapter module during training. Explicit Visual Prompting (EVP) [39] technique incorporates explicit visual prompts to the proposed adapter. ViT-Adapter [8] introduces inductive bias to reconstruct fine-grained multi-scale features. ViT-CoMer [56] performs multi-scale fusion across hierarchical features, which is beneficial for handling dense prediction tasks. Our work explores a novel and effectively adapted backbone for underwater dense prediction tasks.

## 3 VISION TRANSFORMER UNDERWATER-ADAPTER

### 3.1 OVERALL ARCHITECTURE

As illustrated in Figure 3, our ViT-UWA consists of three components: (a) Plain ViT with High-frequency Components Prior (HFCP). (b) Detail Aware Module (DAM). (c) ViT-CNN Interaction Module (VCIM).

For the ViT with HFCP, an input image with the shape of $H \times W \times 3$ and its high-frequency components are first fed into the patch embedding to obtain $16 \times 16$ non-overlapping original image patches and high-frequency components patches, respectively. Then, these patches are added, flattened, and projected to $C$-dimensional feature tokens, and the feature resolution is reduced to 1/16 of the original image. After that, position embedding tokens are added with feature tokens as the input of the first Vision Transformer encoder block.

For the DAM, the image passes through several convolutional neural networks (CNNs) to obtain feature maps $F_1$, $F_2$, $F_3$, and $F_4$ with resolutions of 1/4, 1/8, 1/16 and 1/32. A High-frequency

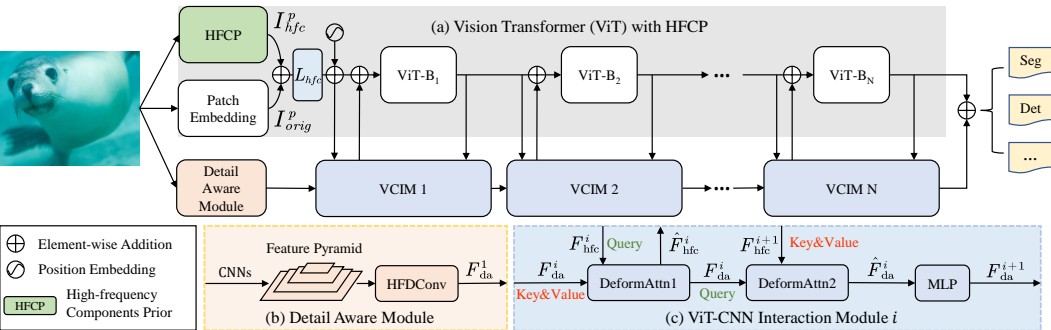

Figure 3: **Overall architecture of ViT-UWA.** ViT-UWA is mainly composed of three components: (a) a plain ViT with high-frequency components prior, whose encoder is divided into $N$ blocks evenly (in Section 3.2). (b) a detail aware module to obtain detail-focused multi-scale features (in Section 3.3). (c) a ViT-CNN interaction module to fuse features of ViT and CNNs (in Section 3.4). In the figure, HFDConv stands for High-frequency Detail Convolution, ViT-B$_i$ stands for the i-th ViT block.

Detail Convolution is used to enhance the detail representation of feature maps and project them to $C$ dimensions. Then, the last three feature maps are flattened and concatenated into feature tokens, as the input for VCIM. The whole process is parallel with the patch embedding of ViT. Given $N$ stage feature interactions, we split the encoder of ViT into $N$ blocks. The high-frequency component feature from the ViT with HFCP and detail-focused feature from DAM interact with each other through multi-scale deformable attention [62]. After $N$-stage feature interactions, the detail-focused multi-scale features from ViT and VCIM are added for underwater dense prediction tasks.

## 3.2 Plain ViT with High-frequency Components Prior

Recent studies [50, 39] have shown that high-frequency information of images like edges, textures, and noise can improve the generalization ability of convolutional neural networks (CNNs), and it is an effective visual prompting for ViT. However, due to the wavelength- and distance-dependent light attenuation and scattering [18], underwater images usually suffer from quality degradation issues such as blurred details, color cast, etc, and lose a large amount of high-frequency information. Therefore, we use the Fourier Transform to recover and capture lost high-frequency information from underwater images.

**High-frequency Components.** As shown in Figure 4(a), for the input image $I$ of shape $H \times W$, we create an all 0 mask $M_0$ with the same shape of $I$. Then we create a square area with all 1 of side length $l = \sqrt{H \times W \times \tau}$ at the center of $M_0$, where $\tau$ indicates the surface ratio of the masked regions. After that, we obtain a binary mask $M \in \{0, 1\}^{H \times W}$. For every pixel in this mask $M_{ij}$, we have:

$$M_{i,j} = \begin{cases} 1, & \text{if } \left| \left( \frac{H}{2} - i \right) \left( \frac{W}{2} - j \right) \right| \leq \frac{HW\tau}{4} \\ 0, & \text{otherwise} \end{cases}. \tag{1}$$

Denoting fft and ifft as the Fast Fourier Transform and its inverse respectively, we have the frequency component $fc = \text{fft}(I)$. We apply $M$ on $fc$ to realize high-pass filtering, then the high-frequency components of $I$ can be computed:

$$I_{hfc} = \text{ifft}\left(fc \cdot (1 - M)\right). \tag{2}$$

We perform the above process on every channel of pixels independently for RGB images.

**High-frequency Components Prior.** As shown in Figure 4(b), after extracting high-frequency components, $I_{hfc}$ is fed into the patch embedding layer to be divided into small patches, denoting $I_{hfc}^p \in {}^c$ and $c = \frac{H}{16} \times \frac{W}{16} \times 3$. Meanwhile, the input image is also fed into the patch embedding layer to obtain $16 \times 16$ non-overlapping original image patches $I_{orig}^p$. By learning a linear layer $L_{hfc}$, $I_{orig}^p$ and $I_{hfc}^p$ are added, flattened and projected into a $C$-dimensional feature $F_{hfc} \in \mathbb{R}^C$. The formula is as follows:

$$F_{\text{hfc}} = L_{hfc}(I_{orig}^p + I_{hfc}^p). \tag{3}$$

Figure 4: **(a) The process to obtain high-frequency components of the input image.** High-frequency components are obtained by fast Fourier transform and its inversion. Both the original image and its high-frequency components are fed into the patch embedding layer. The red-boxed area is set to 0 to highlight the high-frequency information. **(b) The process to add HFCP to the plain ViT.** The patches of the original image and its high-frequency components are flattened, concatenated, and projected to $C$-dimensional feature tokens, as the input of the first ViT block.

## 3.3 DETAIL AWARE MODULE

Recent researches indicate that convolutions enhance transformers' ability to capture local spatial information [43, 55]. And difference convolution (DC) can enhance the representation and generalization capacity of vanilla convolution (VC) [48]. Inspired by these, we design the Detail Aware Module (DAM) to utilize difference convolutions and detail-focused multi-scale features to enhance the high-frequency detail representation of feature maps and structure a detail-focused multi-scale feature pyramid, which can be used in dense prediction tasks.

**High-frequency Detail Convolution.** Difference convolution is typically characterized as the convolution of pixel differences, wherein pixel differences are computed first and then convolved with kernel weights to generate feature maps. Central difference convolution (CDC) and angular difference convolution (ADC) are two typical types of difference convolutions, which optimize computational cost and memory consumption by rearranging learned kernel weights [48]. Due to the complex underwater environment, it is necessary to find accurate boundaries to distinguish different underwater objects for underwater dense prediction tasks. However, due to uneven lighting and low contrast in underwater images, the edges between objects and the waterbody are usually blurred. Difference convolution and its variants have been shown to be effective in tasks that require high-frequency information, such as edge detection [48] and single image dehazing [9]. Considering that a large amount of high-frequency detail information such as edge and texture is lost in underwater images, difference convolution helps enhance the visibility of details by detecting changes in pixel intensity, thus restoring the edges and contours of objects.

Difference convolution can be combined with vanilla convolution to enhance the detail awareness and understanding ability of CNNs [9]. Inspired by this, we propose an adaptive DC and high-frequency detail convolution (HFDConv). For adaptive DC, we first rearrange VC's weight $W \in \mathbb{R}^{C_{\text{out}} \times C_{\text{in}} \times K \times K}$ as a two-dimensional matrix $W' \in \mathbb{R}^{C_{\text{out}} \times C_{\text{in}} \times (K^2)}$, where $C_{\text{in}}$ is the number of input channels, $C_{\text{out}}$ is the number of output channels, $K \times K$ is the size of the convolution kernel. $W'$ are then adjusted for adaptive differencing:

$$W'_{\text{ad}} = W' - \theta \cdot W'[:, :, \text{permute}], \tag{4}$$

where permute is a specific sequence of convolution kernel indexes, like $[3, 0, 1, 6, 4, 2, 7, 8, 5]$. And $\theta$ affecting the high-frequency response of the convolution kernel by decreasing or increasing the influence of specific weight positions. Finally $W'_{\text{ad}}$ are rearranged to the original four-dimensional tensor $W_{\text{ad}} \in \mathbb{R}^{C_{\text{out}} \times C_{\text{in}} \times K \times K}$.

In HFDConv, We employ two convolution layers including one VC and one adaptive DC, deployed in parallel for extracting detail-focused features, which is beneficial to segment blurred edges and detect complicated objects in underwater images. The feature extraction process of HFDConv can be formulated as:

$$\text{HFDConv}(F) = F * (W_{\text{ad}} + W_{\text{vd}}) + (b_{\text{ad}} + b_{\text{vd}}), \tag{5}$$

where $W_{\text{ad}}$, $W_{\text{vd}}$ and $b_{\text{ad}}$, $b_{\text{vd}}$ denote the weights and biases of DC and VC, respectively. $*$ represents the convolution operation. HFDConv not only enhances the model's perception of underwater high-frequency details but also reduces computational costs.

**Detail Aware Feature.** As shown in Figure 3(b), firstly, we use several convolutional neural networks consisting of vanilla stride-2 $3 \times 3$ convolution and BatchNorm [24] to double the number of channels and minimize the size of feature maps. Then, we obtain a feature pyramid $\{F_1, F_2, F_3, F_4\}$ containing 4 feature maps with resolutions of $1/4$, $1/8$, $1/16$ and $1/32$, respectively. After that, four HFDConvs are applied to project these feature maps to $C$ dimensions, which can enhance the detail representation of feature maps and adjust channels to the same as ViT's embedding dimension for feature interaction. At end, we flatten and concatenate the last three detail-focused feature maps into feature tokens $F_{\text{da}}^1 \in \mathbb{R}^{\left(\frac{HW}{8^2} + \frac{HW}{16^2} + \frac{HW}{32^2}\right) \times C}$ named detail aware feature as the input for ViT-CNN interaction module.

## 3.4 ViT-CNN Interaction Module

Due to limitation of single-scale representation and the non-hierarchical feature, the plain ViT does not perform well on underwater dense prediction tasks compared to task-specific methods. The hierarchical feature of CNNs can help solve ViTs' issues on underwater dense prediction like blurred edge segmentation and incorrect category prediction. As a result, inspired by [8], we design a ViT-CNN interaction module (VCIM) to interact DAM's detail-focused multi-scale features with ViT.

As shown in Figure 3(c), VCIM contains two deformable attention blocks and an MLP to conduct multi-scale feature interaction and fusion between DAM and ViT. Firstly, the detail aware feature $F_{\text{da}}^i \in \mathbb{R}^{\left(\frac{HW}{8^2} + \frac{HW}{16^2} + \frac{HW}{32^2}\right) \times C}$ is input as key and value into the $i$-th VICM. The high-frequency component feature of ViT $F_{\text{hfc}}^i \in \mathbb{R}^{\frac{HW}{16^2} \times C}$ serves as the query, and the output feature $\hat{F}_{\text{hfc}}^i$ is obtained through the first multi-scale deformable attention block. All features are normalized by LayerNorm [2]. The formula is as follows:

$$\hat{F}_{\text{hfc}}^i = F_{\text{hfc}}^i + \text{DeformAttn}(F_{\text{hfc}}^i, F_{\text{da}}^i), \tag{6}$$

where $\text{DeformAttn}(\cdot)$ represents multi-scale deformable attention.

In contrast to the above process, we take the detail aware feature $F_{\text{da}}^i$ as a query, and the output $F_{\text{hfc}}^{i+1}$ of the $i$-th ViT block as key and value for the second multi-scale deformable attention block. Then we obtain the next multi-scale detail aware feature $F_{\text{da}}^{i+1} \in \mathbb{R}^{\left(\frac{HW}{8^2} + \frac{HW}{16^2} + \frac{HW}{32^2}\right) \times C}$ through MLP. This feature will serve as the input for the next VCIM. The process can be formulated as:

$$\hat{F}_{\text{da}}^i = F_{\text{da}}^i + \text{DeformAttn}(F_{\text{da}}^i, F_{\text{hfc}}^{i+1}), \tag{7}$$

$$F_{\text{da}}^{i+1} = \hat{F}_{\text{da}}^i + \text{MLP}(\hat{F}_{\text{da}}^i). \tag{8}$$

## 4 Experiments

We select typical tasks in underwater dense prediction: underwater images semantic segmentation, object detection, and instance segmentation, and conduct extensive experiments (with different model sizes, algorithm frameworks, and configurations) on SUIM [26], UIIS [37], and USIS10K [36] datasets, to verify the effectiveness of ViT-UWA. ViT-UWA achieves results that are superior to existing SOTA ViT-based methods (e.g., ViT-Adapter [8], ViT-CoMer [56]) and comparable to task-specific advanced underwater methods (e.g., WaterMask [37], USIS-SAM [36]). In addition, we perform ablation experiments on the proposed modules and qualitative experiments (As shown in Figure 1 and Figure 5, more qualitative comparisons can be found in Appendix A) for underwater dense prediction tasks. These results suggest that ViT-UWA can elevate the performance of plain ViT and serve as a robust backbone for various underwater dense prediction tasks.

### 4.1 Datasets

We conduct experiments on three underwater image datasets: SUIM [26], UIIS [37], and USIS10K [36]. The former is for underwater semantic segmentation and the latter two are for underwater object detection and instance segmentation.

| Method | #Param | #FLOPs | UperNet (IoU) | | | | | | | | mIoU | +MS |
|---|---|---|---|---|---|---|---|---|---|---|---|---|
| | | | BW | HD | PF | WR | RO | RI | FV | SR | | |
| ViT-T [34] | 33.9M | 222G | 83.48 | 61.64 | 16.06 | 37.50 | 59.14 | 55.61 | 44.73 | 55.45 | 51.70 | 53.57 |
| ViT-Adapter-T [8] | 35.9M | 231G | 87.69 | 84.34 | 18.64 | 74.10 | 76.95 | 72.37 | 77.80 | 68.04 | 69.99 | 70.95 |
| ViT-CoMer-T [56] | 40.3M | 231G | 88.86 | 85.44 | 8.34 | 74.40 | 84.65 | 70.02 | 77.95 | 69.05 | 69.84 | 70.34 |
| **ViT-UWA-T (ours)** | 38.2M | 230G | 87.97 | 84.8 | 30.17 | 67.23 | 83.09 | 73.48 | 77.66 | 66.89 | **71.41** | **71.60** |
| ViT-S [34] | 53.5M | 248G | 82.27 | 64.69 | 11.17 | 41.19 | 70.74 | 56.86 | 49.81 | 52.95 | 53.71 | 54.51 |
| Swin-T [40] | 59.8M | 222G | 89.51 | 60.01 | 11.66 | 57.82 | 13.92 | 65.24 | 57.80 | 64.50 | 52.56 | 53.10 |
| RevCol-T [4] | 60.3M | 234G | 89.27 | 88.01 | 21.47 | 74.70 | 82.73 | 74.04 | 83.73 | 69.28 | 72.90 | 73.15 |
| ViT-Adapter-S [8] | 57.5M | 266G | 88.23 | 86.08 | 12.79 | 72.34 | 83.25 | 70.69 | 80.54 | 66.52 | 70.06 | 71.14 |
| ViT-Comer-S [56] | 61.3M | 294G | 88.13 | 87.79 | 15.28 | 75.14 | 84.79 | 71.01 | 80.10 | 69.30 | 71.44 | 72.30 |
| **ViT-UWA-S (ours)** | 62.1M | 265G | 88.05 | 86.19 | 35.71 | 77.61 | 83.85 | 71.20 | 79.12 | 65.86 | **73.45** | **74.35** |
| ViT-B [34] | 126.9M | 339G | 81.15 | 65.26 | 12.92 | 40.88 | 69.97 | 56.39 | 43.13 | 48.68 | 52.30 | 53.49 |
| Swin-B [40] | 121.2M | 296G | 89.44 | 64.43 | 0.99 | 56.62 | 19.05 | 65.54 | 53.68 | 65.08 | 51.85 | 52.71 |
| RevCol-B [4] | 168.8M | 298G | 88.37 | 88.83 | 15.39 | 79.63 | 80.07 | 76.50 | 86.22 | 66.56 | 72.70 | 73.28 |
| ViT-Adapter-B [8] | 133.5M | 375G | 88.44 | 87.15 | 23.21 | 71.41 | 84.99 | 72.53 | 83.56 | 67.82 | 72.39 | 73.19 |
| ViT-Comer-B [56] | 144.6M | 452G | 88.52 | 86.85 | 16.23 | 82.17 | 85.48 | 71.97 | 82.26 | 71.03 | 73.07 | 73.52 |
| **ViT-UWA-B (ours)** | 142.7M | 374G | 88.78 | 86.74 | 34.27 | 81.64 | 83.98 | 72.16 | 83.32 | 71.12 | **75.25** | **75.39** |
| RevCol-L[4] | 306.6M | 418G | 89.48 | 88.38 | 12.72 | 80.56 | 86.88 | 73.04 | 86.67 | 72.08 | 73.73 | 73.94 |
| ViT-Adapter-L[8] | 363.7M | 667G | 89.36 | 88.45 | 17.70 | 79.47 | 87.12 | 72.98 | 85.96 | 71.22 | 74.03 | 74.77 |
| ViT-Comer-L[56] | 426.5M | 1032G | 88.71 | 87.39 | 17.31 | 78.44 | 86.16 | 72.55 | 86.28 | 70.35 | 73.40 | 74.39 |
| **ViT-UWA-L (ours)** | 375.9M | 665G | 88.46 | 87.23 | 46.60 | 77.67 | 84.20 | 76.13 | 83.33 | 71.08 | **76.84** | **77.92** |

Table 1: **Semantic Segmentation on the SUIM.** UperNet [57] are used as segmentation frameworks. "MS" means multi-scale testing. [†] denotes the use of ImageNet-22K pre-training, while the default is to use the regular ImageNet-1K pre-training. The FLOPs are measured with $512 \times 512$ inputs. **BW, HD, PF, WR, RO, RI, FV** and **SR** are 8 categories in the SUIM dataset, representing Waterbody, Human divers, Aquatic plants&sea-grass, Wrecks/ruins, Robots, Reefs&invertebrates, Fish&vertebrates, and Sea-floor&rocks, respectively.

**SUIM Dataset.** The SUIM dataset [26] contains 1525 RGB images for training and validation, with an additional 110 test images provided for benchmark evaluation of semantic segmentation models. The images in the dataset were carefully selected from a large collection of samples gathered during ocean explorations and experiments involving human-robot cooperation. These samples were taken at various locations with different water types. All images of the SUIM dataset are pixel-annotated by human participants.

**UIIS Dataset.** The UIIS dataset [37] is the first large-scale general underwater image instance segmentation dataset. It contains 4628 RGB underwater images, of which 3937 images are used for training/validation and 691 images are used for benchmark evaluation. The images in the dataset are high-quality images carefully selected from approximately 25,000 images, covering multiple application areas such as underwater image enhancement, instance segmentation, and object detection.

**USIS10K Dataset.** The USIS10K dataset [36] is the first large-scale Underwater Image Salient Instance Segmentation dataset, consisting of 10,632 images from various underwater scenes with pixel-level annotations. USIS10K is designed to enhance research in the field of Salient Instance Segmentation (SIS) by including category labels, which aid in detecting semantically dominant regions.

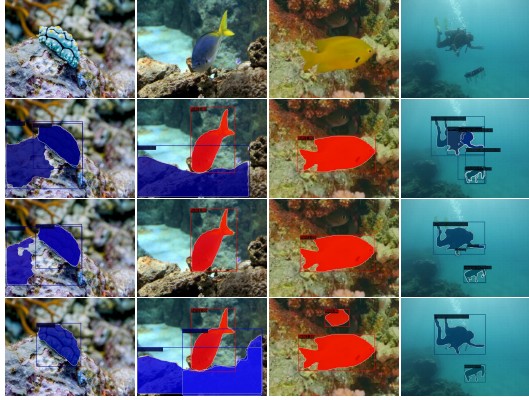

Figure 5: **Qualitative comparison on the USIS10K dataset.** The first row represents the original image, and the second, third and fourth rows represent the results of ViT, ViT-Adapter and ours, respectively.

## 4.2 SEMANTIC SEGMENTATION

**Settings.** Our semantic segmentation experiments are based on MMSegmentation [11] codebase and the SUIM dataset [26]. For easier training, we reformat the SUIM dataset into the Pascal-VOC2012 dataset [13] format. We use UperNet [57] as the basic framework. We follow the same settings of Swin [40] and encompass training for 160K iterations. All experiments are conducted on an NVIDIA 4090 GPU and the batch size is set to 2.

**Comparisons with different backbones.** Table 1 shows the comparisons of both single-scale and multi-scale mIoU between ViT-UWA and various backbones pre-trained on ImageNet-1k and ImageNet-22k, including the plain ViT, vision-specific backbones, and adapted ViT backbones in underwater image semantic segmentation. It shows that, under similar model sizes, our method outperforms other backbones on the SUIM dataset, reaching state-of-the-art performance. For instance, our ViT-UWA-L achieves 76.84% mIoU, outperforming many strong counterparts such as ViT-CoMer-L [56] (+3.44%) and ViT-Adapter-L [8] (+2.81%). These equitable comparisons demonstrate the effectiveness of our ViT-UWA in the underwater image semantic segmentation task. Moreover, our ViT-UWA is more computationally efficient compared to other adapted ViT backbones.

**Comparisons with state-of-the-arts.** In order to further improve the performance, we conduct experiments based on Mask2Former [10], using ViT-UWA as the backbone, and initializing the model with multi-modal pre-training BEiTv2 [44]. As show in Table 2, our ViT-UWA achieves better performance to SOTA methods on the SUIM. For instance, ViT-UWA-L reports a competitive performance of 78.9% mIoU, which is 1.4% higher than ViT-Adapter-G and 2.5% higher than ViT-CoMer-L.

| Method | Backbone | Pre-train | #Param | mIoU | +MS |
|---|---|---|---|---|---|
| UperNet [57] | InternImage-L [51] | IN-22k | 256M | 75.5 | 75.9 |
| Mask2Former [10] | ViT-Adapter-H [8] | BEiT3 [54] | 1.9B | 77.5 | 78.0 |
| Mask2Former [10] | ViT-Adapter-L [8] | BEiTv2 [44] | 571M | 76.6 | 77.2 |
| Mask2Former [10] | ViT-CoMer-L [56] | BEiTv2 [44] | 601M | 76.4 | 77.1 |
| Mask2Former [10] | **ViT-UWA-L** | BEiTv2 [44] | 583M | **78.9** | **79.2** |

Table 2: **Comparisons with previous SOTA for underwater image semantic segmentation.**

## 4.3 OBJECT DETECTION AND INSTANCE SEGMENTATION

| Method | #Param | #FLOPs | UIIS Dataset | | | | | | USIS10K Dataset | | | | | |
|---|---|---|---|---|---|---|---|---|---|---|---|---|---|---|
| | | | $AP^b$ | $AP^b_{50}$ | $AP^b_{75}$ | $AP^m$ | $AP^m_{50}$ | $AP^m_{75}$ | $AP^b$ | $AP^b_{50}$ | $AP^b_{75}$ | $AP^m$ | $AP^m_{50}$ | $AP^m_{75}$ |
| ViT-T [34] | 26M | 223G | 23.9 | 43.6 | 23.6 | 23.4 | 42.6 | 23.4 | 36.8 | 55.1 | 41.9 | 37.1 | 54.9 | 41.6 |
| ViT-Adapter-T [8] | 28M | 260G | 26.2 | 43.7 | 26.2 | 24.7 | 41.8 | 25.4 | 39.9 | 56.0 | 45.0 | 37.6 | 54.9 | 41.6 |
| ViT-CoMer-T [56] | 29M | 262G | 24.7 | 43.8 | 25.4 | 23.7 | 40.9 | 25.2 | 38.5 | 55.0 | 43.2 | 37.4 | 54.1 | 43.0 |
| **ViT-UWA-T (ours)** | 30M | 259G | **26.8** | **45.2** | **26.7** | **25.7** | **44.2** | **27.1** | **40.9** | **56.6** | **46.5** | **38.8** | **56.0** | **44.4** |
| ViT-S [34] | 44M | 329G | 25.1 | 43.0 | 26.3 | 24.7 | 42.8 | 25.9 | 38.8 | 56.2 | 43.2 | 37.4 | 54.1 | 43.0 |
| ViT-Adapter-S [8] | 48M | 401G | 26.4 | 44.3 | 27.3 | 24.4 | 41.8 | 26.0 | 42.3 | 56.8 | 48.4 | 39.1 | 56.2 | 44.7 |
| ViT-CoMer-S [56] | 50M | 407G | 26.0 | 42.8 | **29.1** | 24.2 | 42.7 | 25.1 | 39.7 | 55.9 | 44.4 | 38.0 | 55.3 | 42.9 |
| **ViT-UWA-S (ours)** | 52M | 399G | **28.1** | **45.2** | 28.5 | **26.1** | **43.8** | **27.8** | **43.0** | **57.1** | **49.5** | **39.7** | **56.4** | **45.7** |
| ViT-B [34] | 113M | 690G | 24.9 | 44.7 | 26.2 | 26.3 | 44.2 | 25.4 | 40.9 | 58.1 | 47.7 | 40.2 | 57.9 | 45.4 |
| ViT-Adapter-B [8] | 120M | 830G | 28.2 | 44.7 | 31.0 | 26.1 | 43.4 | 28.3 | 42.1 | 57.1 | 48.0 | 39.8 | 56.9 | 45.0 |
| ViT-CoMer-B [56] | 129M | 877G | 27.0 | 45.1 | 28.9 | 25.4 | 43.8 | 27.6 | 40.5 | 57.0 | 45.9 | 39.2 | 56.0 | 44.5 |
| **ViT-UWA-B (ours)** | 129M | 827G | 29.8 | 46.0 | 32.5 | 26.8 | 44.6 | 29.5 | 44.9 | 58.8 | 51.3 | 42.0 | 58.8 | 47.6 |
| **ViT-UWA-B$^†$ (ours)** | 129M | 827G | **30.9** | **49.1** | **32.8** | **29.0** | **48.6** | **30.9** | **46.4** | **60.8** | **52.3** | **44.2** | **60.0** | **51.2** |

Table 3: **Object detection and instance segmentation with Mask R-CNN on UIIS and USIS10K datasets.** All experiments are conducted with a training schedule 3× (36 epochs). $^†$ denotes the use of ImageNet-22K pre-training, while the default is to use the regular ImageNet-1K pre-training. The FLOPs are measured with 1280×800 inputs.

**Settings.** We utilize the MMDetection [6] codebase to implement our method and conduct object detection and instance segmentation experiments on the UIIS dataset [37] and USIS10K dataset [36]. The object detection and instance segmentation frameworks involve Mask R-CNN [20], Cascade Mask R-CNN [5], ATSS [61], and GFL [33]. We conduct all experiments with a training schedule 3× (36 epochs) on an NVIDIA A100 GPU. We train 2 images on each GPU, using AdamW optimizer with a starting learning rate of 1e-4 and weight decay of 0.05.

**Comparisons with different backbones.** Table 3 presents a comparative analysis of ViT-UWA against various scales of plain ViT and adapted backbones using the UIIS and USIS10K datasets for object detection and instance segmentation. ViT-UWA consistently outperforms competing backbones, particularly in underwater dense prediction tasks. For example, ViT-UWA-B achieves an improvement of +4.9% in box AP and +0.6% in mask AP over plain ViT-B on the UIIS dataset. Similarly, ViT-UWA-B achieves an impressive improvement of +4.0% in box AP and +1.8% in mask AP over ViT-B on the USIS10K dataset. Furthermore, ViT-UWA continues to outperform adapted backbones such as ViT-Adapter [8] and ViT-CoMer [56] across both datasets, highlighting the effectiveness of our approach.

| Method | #Param | #FLOPs | $AP^b$ | $AP^b_{50}$ | $AP^b_{75}$ |
|---|---|---|---|---|---|
| Cascade Mask R-CNN 3x schedule | | | | | |
| ViT-S [34] | 80M | 804G | 29.5 | 44.1 | 30.7 |
| ViT-Adapter-S [8] | 84M | 876G | 30.2 | 44.4 | 31.7 |
| ViT-CoMer-S [56] | 89M | 882G | 29.4 | 44.3 | 30.4 |
| **ViT-UWA-S (ours)** | 89M | 874G | **31.0** | **44.8** | **32.7** |
| ATSS 3x schedule | | | | | |
| ViT-S [34] | 32M | 270G | 27.7 | 43.8 | 28.6 |
| ViT-Adapter-S [8] | 36M | 342G | 27.8 | 43.3 | 28.3 |
| ViT-CoMer-S [56] | 40M | 348G | 28.5 | 43.8 | **30.8** |
| **ViT-UWA-S (ours)** | 40M | 341G | **29.3** | **45.1** | **30.8** |
| GFL 3x schedule | | | | | |
| ViT-S [34] | 32M | 274G | 27.3 | 42.2 | 29.1 |
| ViT-Adapter-S [8] | 36M | 346G | 29.5 | **44.1** | 31.0 |
| ViT-CoMer-S [56] | 40M | 351G | 28.6 | **44.1** | 29.5 |
| **ViT-UWA-S (ours)** | 40M | 344G | **29.7** | 43.7 | **31.3** |

Table 4: **Object detection with different frameworks on the UIIS dataset.**

**Comparisons with different frameworks.** We further evaluate ViT-UWA with different object detection frameworks, the results are shown in Table 4. It can be seen that our approach uniformly outperforms other backbones across various frameworks like Cascade Mask R-CNN [5], ATSS [61], and GFL [33].

**Comparisons with state-of-the-arts.** As show in Table 5, our ViT-UWA-B[†] outperforms the existing SOTA models (initialized with advanced pre-training like EVA-02 [14], DINOv2 [42] and SA-1B [30]) with fewer parameters and only ImageNet-22K pre-training. For example, ViT-UWA-B[†] achieves 1.7% box AP and 2.2% mask AP gains compared to ViTDet-L, which clearly demonstrates the effectiveness of ViT-UWA.

| Method | Backbone | Pre-train | #Param | $AP^b$ | $AP^m$ |
|---|---|---|---|---|---|
| Co-DETR [63] | Swin-L [40] | IN-22K | 218M | 45.5 | - |
| CMask R-CNN [5] | ViTDet-L [35] | EVA-02 [14] | 304M | 44.7 | 42.0 |
| Mask R-CNN [20] | ViT-Adapter-L [8] | DINOv2 [42] | 348M | 42.1 | 41.3 |
| Mask R-CNN [20] | ViT-Adapter-L [8] | IN-22K | 348M | 43.1 | 41.4 |
| Mask R-CNN [20] | SAM-H [30] | SA-1B [30] | 641M | - | 38.5 |
| RSPrompter [7] | SAM-H [30] | SA-1B [30] | 632M | - | 40.2 |
| Mask R-CNN [20] | **ViT-UWA-B**[†] | IN-22K | 129M | **46.4** | **44.2** |

Table 5: **Comparisons with previous SOTA on the USIS10K dataset for underwater image object detection and instance segmentation.**

### 4.4 COMPARISONS WITH TASK-SPECIFIC UNDERWATER METHODS.

**Settings.** We conducted experiments and compared them with advanced task-specific underwater methods, including WaterMask [37] and USIS-SAM [36] for underwater instance segmentation, and SUIM-Net [26] and UISS-Net [22] for underwater semantic segmentation

**Results.** As shown in Table 6, compared with task-specific underwater methods, our method can achieve comparable performance, which demonstrates the great potential of our method in different underwater dense prediction tasks.

| UIIS | $AP^m$ | $AP_{50}$ | $AP_{75}$ | USIS10k | $AP^m$ | $AP_{50}$ | $AP_{75}$ | SUIM | mIoU |
|---|---|---|---|---|---|---|---|---|---|
| WaterMask [37] | 27.2 | 43.7 | 29.3 | WaterMask [37] | 38.7 | 54.9 | 43.2 | SUIM-Net [26] | 53.2 |
| USIS-SAM [36] | **29.0** | 45.4 | **31.5** | USIS-SAM [36] | 43.1 | 59.0 | 48.5 | UISS-Net [22] | 72.1 |
| **ViT-UWA-B**[†] | **29.0** | **48.4** | 30.9 | **ViT-UWA-B**[†] | **44.2** | **60.0** | **51.2** | **ViT-UWA-B** | **75.3** |

Table 6: **Comparisons with task-specific underwater methods of instance segmentation and semantic segmentation on UIIS, USIS10K, and SUIM datasets.**

### 4.5 ABLATION STUDY

**Settings.** We conduct ablation experiments on the ViT-UWA-B, using Mask R-CNN (3×schedule) for underwater image object detection and instance segmentation on the USIS10K dataset. The total

batch size used during the training process is 2, the optimizer employed is AdamW, and the learning rate and weight decay parameters are set to 1e-4 and 0.05 , respectively.

**Ablation for components.** The results of this ablation experiment are shown in Table 7. (1) **DAM**. We verify the effectiveness of DAM by remove DAM from ViT-UWA. With DAM, the model obtains a gain of 1.1 $AP^b$ and 1.3 $AP^m$, which indicates that the DAM helps the model focus on high-frequency detail information in underwater images. (2) **VCIM**. When analyzing the validity of VCIM, we disable the feature interaction and add features from CNNs to the plain ViT directly. With VCIM, ViT-UWA has an improvement of 0.7 $AP^b$ and 0.4 $AP^m$, indicating that bidirectional feature interaction is

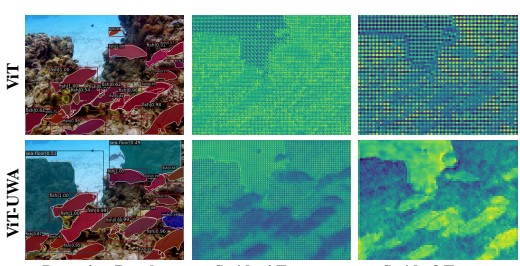

Figure 6: **Visualization of feature maps for object detection and instance segmentation.**

beneficial for dense prediction. (3) **HFCP**. We evaluate the effectiveness of the HFCP by replacing it by other underwater imagery restoration methods with light architectures. After replacing, the model will utilize USUIR [16] to recover degraded underwater images. After the replacement, the model's $AP^b$ and $AP^m$ decrease by 2.3 and 0.6 AP. Compared to visually recovering underwater images, HFCP can better recover high-frequency information that is of greater interest for dense prediction tasks such as segmentation and detection.

In addition, we also visualize the stride-4 and stride-8 feature maps in Figure 6, which shows that the features of our ViT-UWA are more fine-grained and have more high-frequency detail information like edges and textures, further validating the validity of our components.

| Methods | $AP^b$ | $AP^m$ |
|---|---|---|
| ViT-UWA | **44.9** | **42.0** |
| w/o DAM | 43.8 (-1.1) | 40.7 (-1.3) |
| w/o VCIM | 44.1 (-0.7) | 41.6 (-0.4) |
| replace HFCP | 42.6 (-2.3) | 41.4 (-0.6) |

Table 7: **Ablation for components.**

| $N$ | $AP^b$ | $AP^m$ | #Param |
|---|---|---|---|
| 1 | 42.8 | 41.1 | 117M |
| 2 | 43.4 | 41.5 | 121M |
| 4 | **44.9** | **42.0** | 129M |
| 6 | 44.8 | 41.8 | 137M |

Table 8: **Number of ViT-CNN interaction.**

**Number of ViT-CNN interaction.** In Table 8, we analyze the influence of the number of ViT-CNN interaction modules. We find that as N increases, the model performance reaches a plateau, and introducing more interaction modules does not consistently improve performance. Consequently, we set N to 4 as a standard.

**Different mask ratio of HFCP.** Table 9 illustrates the influence of the varying mask ratio of HFCP. The larger the mask ratio, the darker the high-frequency component of the image, that is, the less high-frequency information extracted. Simultaneously, We observe that $AP^b$ and $AP^m$ peak when $\tau = 0.25$ and the performance decreases with the increase of mask ratio. Therefore, we adopt $\tau = 0.25$ as the default setting.

| $\tau$ | $AP^b$ | $AP^m$ |
|---|---|---|
| 0.1 | 44.2 | 41.6 |
| 0.25 | **44.9** | **42.0** |
| 0.5 | 44.5 | 41.8 |
| 1 | 44.1 | 41.3 |
| 2 | 43.5 | 40.9 |

Table 9: **Different mask ratio of HFCP.** The model performs best when $\tau = 0.25$.

## 5 CONCLUSION

In this work, we propose ViT-UWA, a detail-focused and adapted ViT backbone for underwater dense prediction tasks. Without altering the original ViT architecture, we introduce high-frequency components prior to the plain ViT and improve ViT's perception of underwater high-frequency details by a detail aware module. Our method effectively solves the issues such as incorrect category prediction and blurred segmented edges faced by ViT in underwater dense prediction tasks. Extensive experiments on semantic segmentation, instance segmentation, and object detection for underwater imagery show that our method can achieves comparable or superior performance compared to both plain and adapted ViT backbones, as well as task-specific underwater methods.

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

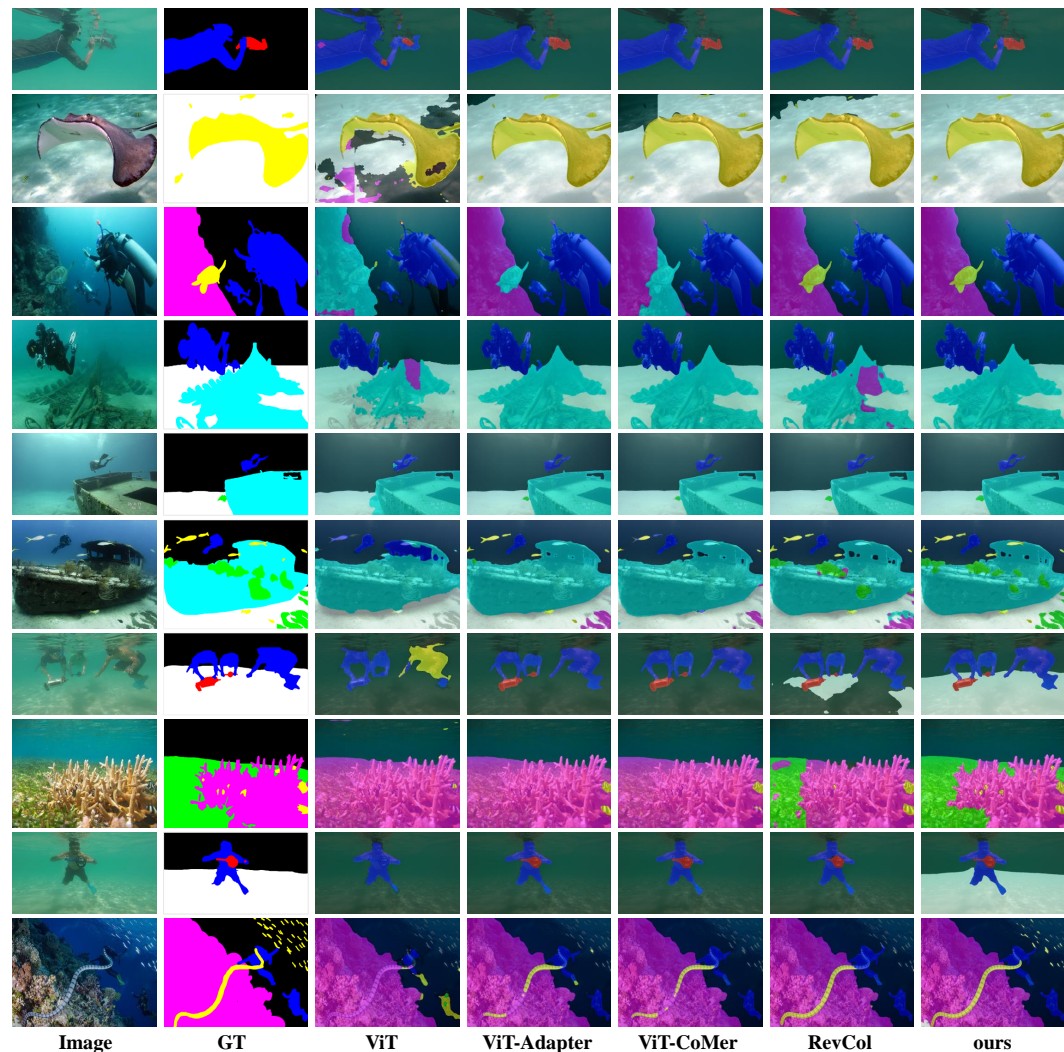

Figure 7: **More qualitative comparison on the SUIM dataset for underwater semantic segmentation.**

# A    MORE QUALITATIVE COMPARISON

## A.1    SEMANTIC SEGMENTATION

We show more qualitative comparisons on the SUIM dataset for underwater semantic segmentation in Figure 7 to demonstrate the effectiveness of our ViT-UWA. It can be seen that compared to adapted ViT Backbone such as ViT-Adapter [8] and ViT-CoMer [56], benefiting from our High-frequency Components Prior, ViT-UWA can better recover lost underwater high-frequency information like edges and segment the accurate semantic boundaries (as shown in Figure 7, rows 6 and 8).

## A.2    OBJECT DETECTION AND INSTANCE SEGMENTATION

We also present more visual comparisons on the USIS10K dataset for underwater object detection and instance segmentation in Figure 8. It can be seen that ViT-UWA can enhance the perception of high-frequency details and detect objects more accurately (as shown in Figure 8, rows 3 and 7) due to the effectiveness of Detail Aware Module.

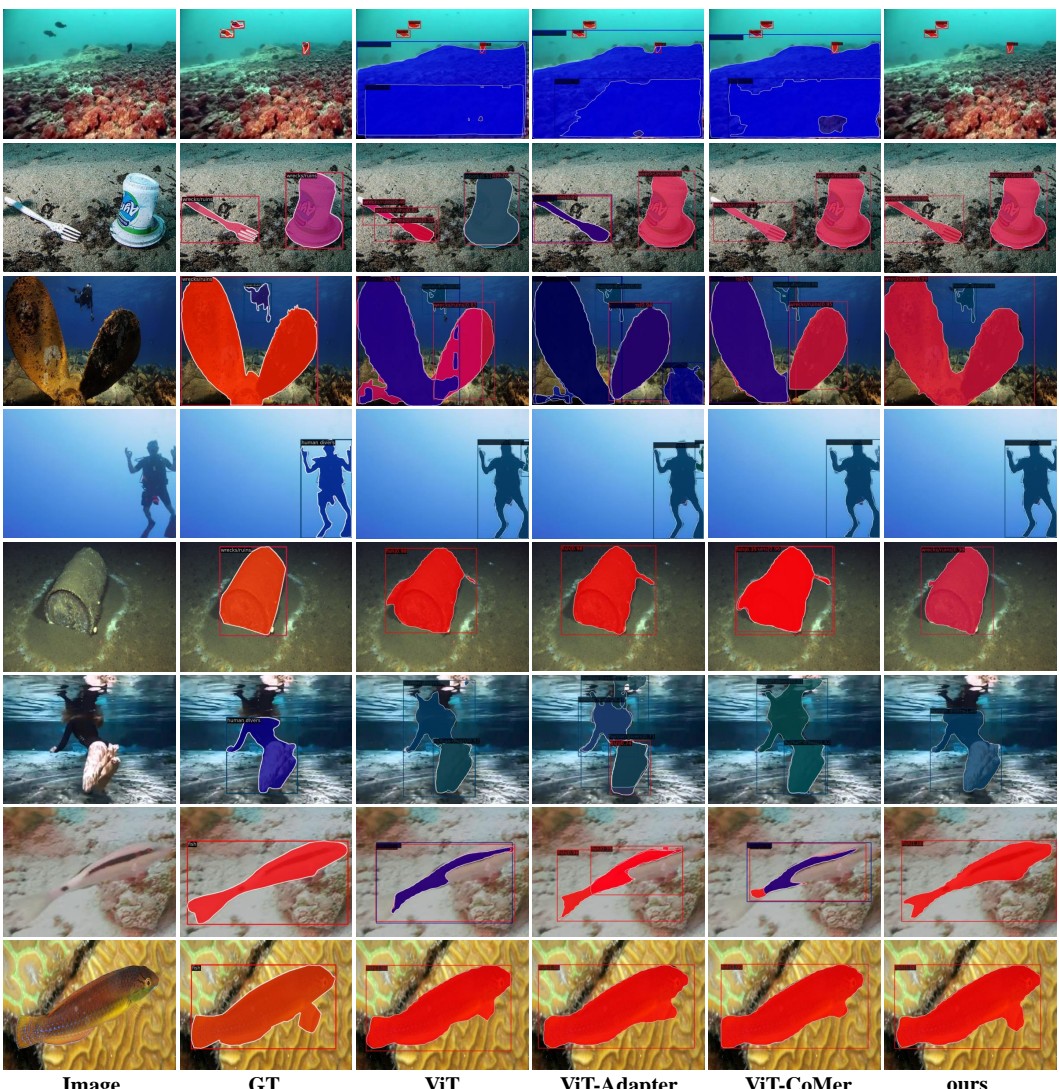

| Image | GT | ViT | ViT-Adapter | ViT-CoMer | ours |

Figure 8: **More qualitative comparison on the USIS10K dataset for underwater object detection and instance segmentation.**

## B    MORE ABLATION EXPERIMENTS

### B.1    GENERALIZATION ABILITY OF VIT-UWA

To verify the generalisation ability of ViT-UWA on natural images, we retrained ViT-UWA-T on the COCO dataset [38]. For fair comparison, we set batch size to 16 and use Mask R-CNN with a training schedule 1× (12 epochs). As shown in Table 10, compared with the plain ViT, vision-specific methods, and adapted ViT backbones, our ViT-UWA achieves comparable performance with similar model size. As some modules are designed for underwater images, ViT-UWA's performance on natural images is somewhat compromised.

| Method | Pre-train | #Param | AP$^b$ | AP$^m$ |
|---|---|---|---|---|
| PVT-T [53] | IN-1K | 33M | 36.7 | 35.1 |
| PVTv2-B1 [52] | IN-1K | 34M | 41.8 | **38.8** |
| ViTDet-T [35] | IN-1K | 26M | 33.5 | 35.7 |
| ViT-T [34] | IN-1K | 27M | 35.5 | 33.5 |
| ViT-Adapter-T [8] | IN-1K | 28M | 41.1 | 37.5 |
| ViT-CoMer-T [56] | IN-1K | 29M | **42.1** | 38.0 |
| **ViT-UWA-T** | IN-1K | 30M | 41.8 | 38.1 |

Table 10: **Object detection and instance segmentation with Mask R-CNN on COCO val2017.**

## B.2 HIGH-FREQUENCY DETAIL CONVOLUTION

We show the results of the following ablation experiments about the type of difference convolution of the High-frequency Detail Convolution in Table 11. We replace the adaptive difference convolution with central difference convolution (CDC), angular difference convolution (ADC), and their combination to evaluate the effectiveness of the adaptive DC. The model gives the best performance when using adaptive DC. Notably, it can be replaced by other more advanced and efficient difference convolutions to further improve performance in the future.

| DC | $AP^b$ | $AP^m$ | #Param |
|---|---|---|---|
| adaptive DC | **44.9** | **42.0** | 129M |
| CDC | 43.5 (-1.4) | 40.8 (-1.1) | 129M |
| ADC | 43.9 (-1.0) | 41.2 (-0.8) | 129M |
| CDC + ADC | 44.1 (-0.8) | 41.4 (-0.6) | 134M |

Table 11: **Type of difference convolution.**

## B.3 DIFFERENT ATTENTION MECHANISMS

To explore the effect of attention mechanism on the model, we adopt ViT-UWA-T as the basic model and study 3 different attention mechanisms in ViT-CNN interaction module. As shown in Table 12, compared with ordinary cross attention [49] with quadratic complexity, deformable Attention [62] with linear complexity can result in fewer parameters, faster computation, and better performance.

| Attention Mechanism | Complexity | #Param | #FLOPs | $AP^b$ | $AP^m$ |
|---|---|---|---|---|---|
| Cross Attention [49] | Quadratic | 32M | 695G | 40.1 | 38.2 |
| Efficient Attention [46] | Linear | 31M | 290G | 40.5 | 38.4 |
| Deformable Attention [62] | Linear | 30M | 259G | **40.9** | **38.8** |

Table 12: **Ablation of different attention mechanisms.**

## B.4 DETAIL AWARE MODULE

To verify the effectiveness of our Detail Aware Module (DAM), we replace DAM with simple CNN structures borrowed from ResNet [19] and similar-function modules from ViT-Adapter [8], ViT-CoMer [56], and InternImage [51] to construct multi-scale features in ViT-UWA-T. As shown in Table 13, under a similar scale of parameters, our method achieves the lowest computational cost and the best performance, indicating that DAM can obtain multi-scale features with rich high-frequency details more efficiently.

| Method | #Param | #FLOPs | $AP^b$ | $AP^m$ |
|---|---|---|---|---|
| DAM (ours) | 30M | 259G | **40.9** | **38.8** |
| CNN [19] | 28M | 260G | 38.9 | 37.0 |
| SPM [8] | 29M | 261G | 40.3 | 38.1 |
| MRFP [56] | 30M | 264G | 39.2 | 37.9 |
| Stem + DS [51] | 31M | 268G | 39.6 | 38.2 |

Table 13: **Ablation of Detail Aware Module.** "DS" means downsampling layers.

## B.5 DIFFERENT METHODS OF UNDERWATER ENHANCEMENT.

To investigate the impact of High-frequency Components Prior (HFCP) and different underwater enhancement methods on the model, we removed HFCP and trained the model with enhanced underwater images. We utilized various open-source underwater image enhancement methods from recent years (e.g., NU$^2$Net [18], PUIE-Net [15]) to enhance the training set of USIS10K and evaluated the trained models using the original test set. Table 14 shows the results of this ablation experiment, we observed that when training with enhanced images, the

| Method | Enhancement | Training strategy | $AP^b$ | $AP^m$ |
|---|---|---|---|---|
| ViT-UWA (Full Model) | - | End-to-End | **44.9** | **42.0** |
| ViT-UWA (w/o HFCP) | - | End-to-End | 42.9 | 40.0 |
| ViT-UWA (w/o HFCP) | FUnIE [25] | Enhance-then-Train | 38.1 | 35.2 |
| ViT-UWA (w/o HFCP) | NU$^2$Net [18] | Enhance-then-Train | 37.3 | 35.3 |
| ViT-UWA (w/o HFCP) | PUIE-Net [15] | Enhance-then-Train | 34.7 | 31.6 |

Table 14: **Different methods of underwater enhancement.** NU$^2$Net and FUnIE are supervised underwater image enhancement methods, while PUIE-Net based on distribution estimation and consistency.

model's performance experienced a certain degree of degradation, and similar conclusions were also reported in [59, 36]. This may be due to underwater image enhancement methods altering the feature distribution of underwater images and introducing additional noise (e.g., halo effect), which negatively impacts dense prediction tasks. Moreover, we compare the visualization results of feature maps between the original model and the model without HFCP in Figure 9 and Figure 10. It can be seen that with HFCP, the model can recover more high-frequency information, such as finer boundaries and richer details.

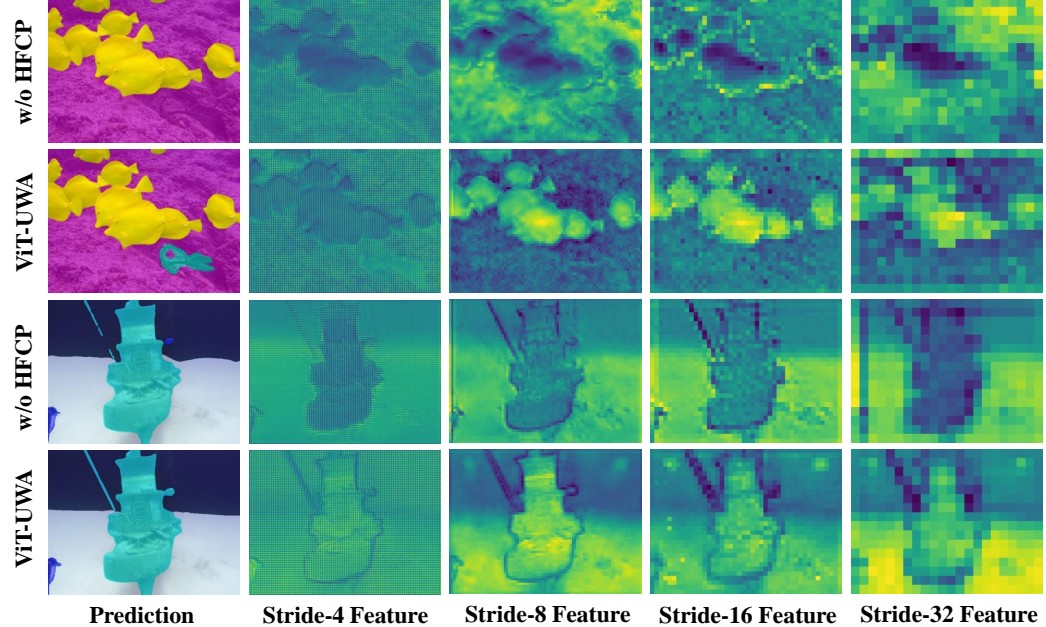

Figure 9: **More qualitative comparison of feature maps for underwater semantic segmentation.** With High-frequency Components Prior (HFCP), our ViT-UWA can capture more high-frequency information (e.g., edges and textures), resulting in feature maps with sharper and more defined edges.

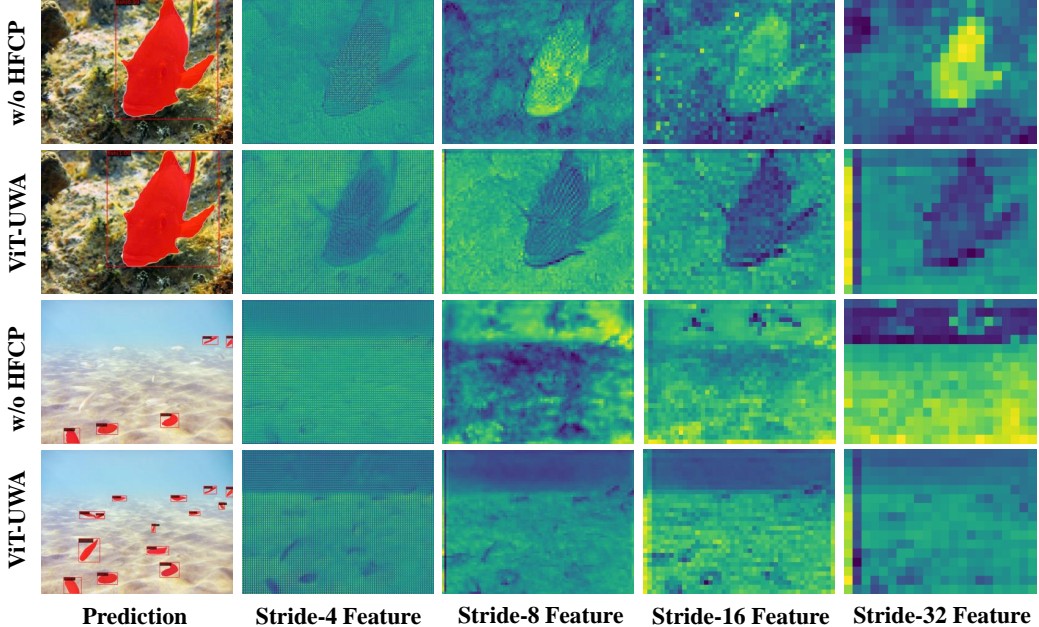

Figure 10: **More qualitative comparison of feature maps for underwater object detection and instance segmentation.** With High-frequency Components Prior (HFCP), our ViT-UWA can obtain multi-scale features with richer details and textures.

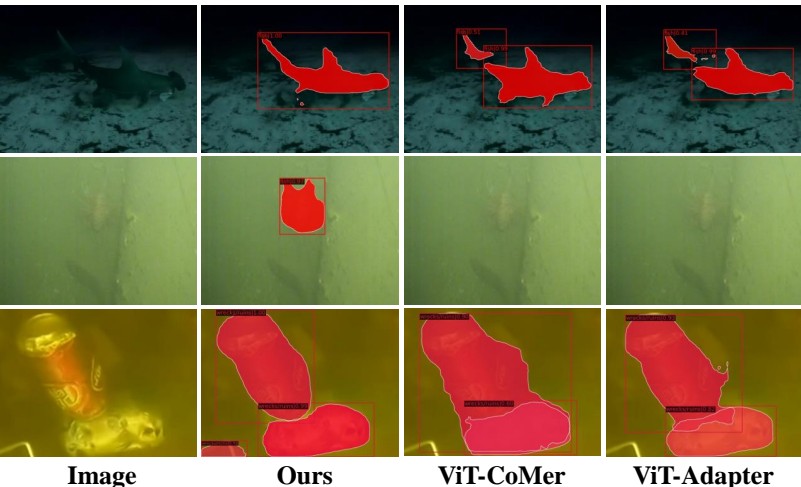

| **Image** | **Ours** | **ViT-CoMer** | **ViT-Adapter** |

Figure 11: **Qualitative comparison in challenging real-world underwater application scenarios.**

### B.6 REAL-WORLD EFFICIENCY ANALYSIS.

We evaluated key metrics in Table 15 that are critical for real-world applications, including FLOPs and inference time. Our ViT-UWA achieved minimal computational overhead and relatively fast inference speed. Moreover, we conducted qualitative comparison in some challenging real-world underwater application sce-

|  | ViT-UWA-B | ViT-CoMer-B | ViT-Adapter-B |
|---|---|---|---|
| FLOPs | **827G** | 877G | 830G |
| inference time | 83.3ms | 98.0ms | **76.34ms** |

Table 15: **Comparison of FLOPs and inference time.**

narios. As shown in Figure 11, in low-light and turbid underwater environments, other ViT-based methods often encounter issues such as errors in object count detection and detection failures. In contrast, our ViT-UWA can alleviate these issues to some extent. This demonstrates the significant potential and value of our ViT-UWA in real-world applications.

