# OpenReview forum: "ViT-UWA: Vision Transformer Underwater-Adapter for Dense Predictions Beneath the Water Surface"
_ICLR.cc/2025/Conference — Submitted to ICLR 2025_

### Official Review · Reviewer_JkrF · 2024-10-20

**Soundness:** 3
**Presentation:** 3
**Contribution:** 3
**Rating:** 6
**Confidence:** 5

**Summary:**

This paper addresses the dense prediction tasks (semantic segmentation, instance segmentation and object detection) from underwater images via an adapted Vision Transformer (ViT) model. The work (named ViT-UWA) adds the High-Frequency Component Prior (HFCP) into the tokens for the plain ViT, and designs two modules of Detail Aware Module (DAM) and ViT-CNN Interaction Module (VCIM), where DAM is devised to extract detail-focused multi-scale features for VCIM and VCIM interacts with the plain ViT to fuse and refine the features for dense predictions. The experiments were conducted on three underwater image datasets, covering semantic segmentation, instance segmentation and object detection, compared with existing general ViT adapters and underwater dense prediction methods.

**Strengths:**

1. The effectiveness and efficiency of the proposed method seems good compared to the related works (Figure 2).
2. The proposed method ViT-UWA has advantages to adapt the underwater images with specific degradation on the details (Figure 3).

**Weaknesses:**

1. While the proposed method achieves good performance, it seems that the claimed contributions, i.e., HFCP, DAM and VCIM, are designed by considering some related works, thus it is somewhat questionable about the novelty of the design.
2. For efficiency evaluation, it's better to consider the aspects of Parameters, FLOPs, as well as training and inference time, especially the FLOPs and inference time for real-word applications.
3. The design seems to only consider the detail loss (HFCB and DAM) on underwater images, while the other degradations caused by the light absorption and scattering on underwater images are not considered, so how does the model learn these different degradations for dense prediction tasks.

**Questions:**

1. What are the specific differences on each contribution compared to existing similar works?
2. Please consider to evaluate the efficiency towards real-world applications.
3. How does the proposed model learn different degradations on underwater images for dense prediction tasks?
4. Please provide the discussion about the SAM 2 and proposed ViT-UWA.

---

> ### Author Response · Authors · 2024-11-20
> **Response to Reviewer JkrF**
>
> Thank you for your insightful comments and valuable suggestions. We have addressed each of your questions as follows:
>
> **Regarding Weakness 1 and Question 1:**
>
> We propose an adapted ViT backbone to address underwater dense prediction tasks in a unified manner.
>
>  We designed HFCP for ViT architecture to recover high-frequency information lost in underwater images. The original ViT focuses more on global information through the self-attention mechanism. Compared to other adapted ViT backbones that do not alter the plain ViT architecture, we observed that using the high-frequency components of underwater images as a prior for ViT effectively helps it capture underwater local high-frequency information. Moreover, compared to directly enhancing the images (Appendix B.5), our HFCP is better suited for underwater dense prediction tasks.
>
>  Convolutions can enhance transformers' ability to capture local spatial information, while differential convolutions can enhance the representation and generalization capacity of convolutions. We proposed HFDConv by combining adaptive differential convolutions (adaptive DC) and standard convolutions to enhance the visibility of details by detecting changes in pixel intensity (for detailed technical information, please refer to Section 3.3). Through using CNNs and HFDConv in DAM, we obtained multi-scale features focused on fine details, which are highly beneficial for underwater dense prediction tasks.
>
>  Our VCIM acts as a bridge between ViT and CNN, enabling bidirectional feature fusion through efficient deformable attention. It integrates the high-frequency features restored by HFCP with the multi-scale features captured by DAM, further enhancing the model's performance in underwater dense prediction tasks.
>
> **Regarding Weakness 2 and Question 2:**
>
>  We have provided the FLOPs results in Tables 1, 3, and 4 in the main text. We also tested the inference time of our method and other adapted ViT methods, and the results are shown in the following Table 1 (also shown in Table 15 of Appendix B.6). Our ViT-UWA achieved minimal computational overhead and relatively fast inference speed. Moreover, we conducted qualitative comparison in some challenging real-world underwater application scenarios. As shown in Figure 11 of Appendix B.6, in low-light and turbid underwater environments, other ViT-based methods often encounter issues such as errors in object count detection and detection failures. In contrast, our ViT-UWA can alleviate these issues to some extent.
>
>  ||ViT-UWA-B|ViT-CoMer-B|ViT-Adapter-B|
>  |-|-|-|-|
>  |FLOPs|**827G**|877G|830G|
>  |inference time|83.3ms|98.0ms|**76.34ms**|
>
>  **Table 1:** Comparison of FLOPs and inference time.
>
> **Regarding Weakness 3 and Question 3:**
>
> There are many factors that contribute to underwater image degradation, such as light absorption and scattering. Underwater image degradation involves the loss of high-frequency information, color shifts, low contrast, etc. Based on our early attempts and analysis of related literature [1], using image enhancement methods (e.g., color correction and contrast improvement) to mitigate degradation totally does not improve the performance of subsequent tasks; in some cases, it may even lead to performance degradation.
>
> From another side, our ViT-UWA focuses on dense prediction tasks, including detection and segmentation. Therefore, we prioritize the recovery of high-frequency details that are beneficial for improving our model's performance. During experiments, we found that our HFCP module could leverage frequency-domain information to mitigate underwater degradation issues such as color shifts and low contrast, thereby improving the model's performance. For instance, color shifts (evident in Rows 7 and 9 of Figure 7 in the paper) blur the boundaries between water bodies and the seafloor. Compared to other methods, our model can clearly distinguish these boundaries. Similarly, low contrast (illustrated in Row 7 of Figure 8 in the paper) makes it challenging for ViT-based methods to differentiate between fish and the background. In contrast, our model is able to effectively detect fish camouflaged within the background.
>
> [1] Xu S, Zhang M, Song W, et al. A systematic review and analysis of deep learning-based underwater object detection[J]. Neurocomputing, 2023, 527: 204-232.
>
> **Due to the character limit of a single comment, we will address Question 4 in the next comment.**

---

> ### Author Response · Authors · 2024-11-20
>
> **Regarding Question 4:**
>
>  SAM2 relies on prompts to leverage its segmentation capabilities. Our ViT-UWA is an underwater dense prediction model based on Vision Transformer, which does not rely on any prompts and achieves outstanding performance in underwater image dense prediction tasks. We compared the performance of our method and SAM2 in underwater instance segmentation [2]. The experimental results are shown in the Table 2. ViT-UWA-B† achieves 11.1% mask AP and 16.2% mask AP gains compared to SAM2-H on UIIS and USIS10K.
>
>  |||UIIS||||USIS||||
>  |-|-|-|-|-|-|-|-|-|-
>  |Method|Backbone|AP(m)|AP50|AP75| |AP(m)|AP50|AP75|
>  |ViT-UWA-B†|ViT-B|**29.0**|**48.6**|**30.9**| |**44.2**|**60.0**|**51.2**|
>  |SAM2|Hiera-Tiny|7.2|9.9|8.2| |18.0|21.7|20.0|
>  |SAM2|Hiera-Base+|15.7|20.6|18.0| | 30.3|36.5|34.1|
>  |SAM2|Hiera-Large|17.9|23.8|19.7| | 28.0|33.1|31.1|
>
>  **Table 2:** Comparison of SAM2 and ViT-UWA on UIIS and USIS10K datasets.
>
> [2] Lian S, Li H. Evaluation of segment anything model 2: The role of sam2 in the underwater environment[J]. arXiv preprint arXiv:2408.02924, 2024.

---

### Official Review · Reviewer_nQpN · 2024-10-21

**Soundness:** 2
**Presentation:** 3
**Contribution:** 1
**Rating:** 5
**Confidence:** 4

**Summary:**

This paper introduces a new backbone, ViT-UWA, specifically designed for underwater vision tasks. ViT-UWA consists of two branches: a ViT branch and a convolutional branch, which leverage the ViT-CNN Interaction Module (VCIM) to facilitate information sharing between them. Moreover, the ViT branch incorporates high-frequency components as additional input, while the CNN branch utilizes High-frequency Detail Convolution (HFDConv) to focus details. The model has achieved good results in several underwater tasks, such as object detection and instance segmentation.

**Strengths:**

1. The paper is well written, and the methods are introduced very clearly.
2. The paper conducted experiments on many tasks, demonstrating the performance advantages of ViT-UWA.

**Weaknesses:**

1. The biggest drawback of this paper is the lack of originality; the proposed ViT-UWA is composed of existing module combinations, such as the adaptive DC in HFDConv and the Deform Attention in VCIM. The so-called HFCP is merely the concatenation of the high-pass filtered result of the original image with the original image input to the model. Therefore, I believe the contribution of this paper to the community is insufficient.
2. Although ViT-UWA is specifically designed for underwater tasks, some unique designs, such as high-frequency components and detail information, are important in any scenario. However, ViT-UWA performs mediocrely on the Coco dataset in Table 10, while it performs well in underwater scenes. The authors' motivation for the model design does not convince me, and the reason for ViT-UWA's better performance in underwater scenarios remains unclear.

**Questions:**

1. Should the phrase "... improve by 2.3 and 0.6 AP" in line 504 be changed to "decrease by 2.3 and 0.6 AP"?
2. What would happen if HFCP were directly removed from Table 7 instead of being replaced by USUIR? Additionally, have any better supervised underwater image enhancement methods been tried instead of the unsupervised USUIR method, which often performs worse than supervised methods in many scenarios?
3. Why does VCIM use Deform Attention instead of ordinary cross attention?

---

> ### Author Response · Authors · 2024-11-20
> **Response to Reviewer nQpN**
>
> Thank you for your comments. We have addressed each of your questions as follows:
>
> **Regarding Weakness 1:**
>
> We propose an adapted ViT backbone to address underwater dense prediction tasks in a unified manner.
>
> We developed HFCP specifically for the ViT architecture to address the challenge of recovering high-frequency information lost in underwater images. While the original ViT architecture emphasizes global information via the self-attention mechanism, our HFCP leverages the high-frequency components of underwater images as a prior, effectively enabling ViT to capture local high-frequency details. Compared to other adapted ViT backbones that retain the plain ViT architecture, our approach significantly enhances performance in underwater scenarios. Additionally, as demonstrated in Appendix B.5, HFCP is more suitable for underwater dense prediction tasks than direct image enhancement methods.
>
> To further enhance the model's capability, we introduced HFDConv, which combines adaptive differential convolutions (adaptive DC) with standard convolutions. This design strengthens the ability of the model to represent and generalize fine details by detecting pixel intensity changes (detailed in Section 3.3). Integrated into DAM, HFDConv enables the extraction of multi-scale features focused on fine-grained details, providing substantial benefits for underwater dense prediction tasks.
>
> Lastly, our VCIM bridges ViT and CNN by facilitating bidirectional feature fusion through deformable attention. It effectively combines the high-frequency features restored by HFCP with the multi-scale features extracted by DAM, resulting in improved model performance for underwater dense prediction tasks.
>
> **Regarding Weakness 2:**
>
> Due to wavelength- and distance-dependent light attenuation and scattering, underwater images commonly suffer from significant loss of high-frequency information. This phenomenon is rare in terrestrial scenes and typically occurs only under specific conditions (e.g., foggy or low-light). Compared to terrestrial images, underwater images have a greater demand of high-frequency information. Therefore, we introduced high-frequency prior into the model to alleviate this issue.
>
> On the COCO dataset, our ViT-UWA-T achieved the second-highest box AP and mask AP scores among state-of-the-art models with similar parameter sizes. In future work, we will try to adapt ViT-UWA to other complex scenarios to further validate the generalization capabilities of our model according to your advice.
>
> **Regarding Question 1:**
>
> Thank you very much for your careful review and valuable feedback. We sincerely apologize for the oversight in line 504. The correct phrasing should indeed be “decrease by 2.3 and 0.6 AP”. We have corrected the statement in the revised manuscript to clarify the performance change as you suggested.
>
> **Regarding Question 2:**
>
> We have conducted additional experiments to address your concerns. Specifically, we have removed HFCP entirely and replaced it with some supervised underwater image enhancement methods. The results of these experiments are presented in Table 14 of Appendix B.5 (also shown in the following Table 1), along with an analysis of their impact on performance. we observed that when training with enhanced images, the model's performance experienced a certain degree of decline. This may be due to underwater image enhancement methods altering the feature distribution of underwater images and introducing additional noise (e.g., halo effect), which negatively impacts dense prediction tasks. Additionally, we have included comparative visualizations (Figures 9 and 10 in Appendix B.5) to illustrate the differences in feature maps after removing HFCP. These additions highlight the effectiveness of HFCP.
>
>  |Method|Enhancement|Training Strategy|AP(b)|AP(m)|
>  |-|-|-|-|-|
>  |ViT-UWA (Full Model)|-|End-to-End|**44.9**|**42.0**|
>  |ViT-UWA (w/o HFCP)|-|End-to-End|42.9|40.0|
>  |ViT-UWA (w/o HFCP)|FUnIE| Enhance-then-Train|38.1|35.2|
>  |ViT-UWA (w/o HFCP)|NU2Net| Enhance-then-Train|37.3|35.3|
>  |ViT-UWA (w/o HFCP)|PUIE-Net| Enhance-then-Train|34.7|31.6|
>
>  **Table 1:** Different methods of underwater enhancement.
>
> **Regarding Question 3:**
>
> We have added an ablation study on the attention mechanism in Table 12 of Appendix B.3 (also shown in the following Table 2). Ordinary cross attention has quadratic complexity, which significantly increases the model's computational cost and training time. However, sparse attention like deformable attention has linear complexity, which reduces the model's computational cost and improves performance to some extent.
>
>  |Attention Mechanism|Complexity|#Param|#FLOPs|AP(b)|AP(m)|
>  |-|-|-|-|-|-|
>  |Cross Attention|Quadratic|32M|695G|40.1|38.2|
>  |Efficient Attention|Linear|31M|290G|40.5|38.4|
>  |Deformable Attention|Linear|30M|259G|**40.9**|**38.8**|
>
>  **Table 2:** Ablation of different attention mechanisms.

---

> > ### Comment · Reviewer_nQpN · 2024-11-23
> > **Response to Authors**
> >
> > Thanks for the authors' response. Some errors are corrected, some questions are addressed, and the experiments are more sufficient. But I still think the biggest drawback of the paper is novelty and it is a boardline paper with enough experiments and SOTA performance. So I raise my score from 3 to 5.

---

> > > ### Author Response · Authors · 2024-11-26
> > > **Response to Reviewer nQpN**
> > >
> > > Thank you for your positive response—we deeply appreciate your willingness to reconsider your score. Regarding the novelty of our work, we provide more explanations below:
> > >
> > > First, to the best of our knowledge, this is the first time a model has been proposed for underwater scenes to address multiple underwater image dense prediction tasks (i.e., semantic segmentation, object detection, and instance segmentation). Dense prediction is valuable in underwater vision scenarios, such as visually guided underwater robots, marine habitat mapping, and underwater target detection.
> > >
> > > Then, in HFCP, we utilized the fast Fourier transform and its inverse transform to extract the high-frequency flux of underwater images. The original image and its high-frequency components are processed separately and then fused into a new feature representation through a linear transformation, effectively restoring high-frequency information in underwater images. Unlike other ViT-based methods, HFCP introduces high-frequency components as prior information, enabling the ViT model to focus not only on low-frequency information (representing the overall structure of the image) but also on the high-frequency details that are often lost in underwater environments. HFCP contributes to a 2% improvement in both box AP and mask AP for our model, which is a relatively significant enhancement. For reference, FreqFusion [1] (TPAMI 2024) achieved a 1.6% improvement in box AP and a 1.4% improvement in mask AP for state-of-the-art models on the COCO dataset.
> > >
> > > Underwater images often suffer from quality degradation issues such as color distortion and low contrast, making it difficult to distinguish between objects and their backgrounds. Compared to traditional convolutions, differential convolutions (DC) enhance detail perception by focusing on differences between pixels. Our proposed adaptive DC introduces an adaptive differential adjustment to the convolution kernel, modifying the high-frequency response by selectively increasing or decreasing the influence of specific weight positions. By combining adaptive DC with standard convolutions in HFDConv, more high-frequency details can be detected and restored, enabling the model to better differentiate objects from water regions. Compared to the classic Central Differential Convolution (CDC), our adaptive DC achieved a 1.4% improvement in box AP and a 1.1% improvement in mask AP, demonstrating its effectiveness in addressing underwater image challenges.
> > >
> > > [1] Chen L, Fu Y, Gu L, et al. Frequency-aware feature fusion for dense image prediction[J]. IEEE Transactions on Pattern Analysis and Machine Intelligence, 2024.

---

### Official Review · Reviewer_ZvDc · 2024-11-02

**Soundness:** 2
**Presentation:** 3
**Contribution:** 2
**Rating:** 6
**Confidence:** 4

**Summary:**

In this paper, a novel underwater dense prediction backbone by combining the plain ViT. Besides, the high-frequency components prior and the detail aware module are proposed. The proposed method was trained and tested on multiple datasets and achieved comparable results. Finally, multiple ablation experiments illustrate the effectiveness of high-frequency components prior and the detail aware module in this method.

**Strengths:**

1. In this paper, multiple experiments were conducted on different datasets and compared quantitatively and qualitatively with multiple methods.
2. This paper is generally smooth, and the paper is easy to understand.
3. The visualization results of the method proposed in the paper look better than other compared methods.

**Weaknesses:**

1.	In ablation experiments, only indicators show whether the high-frequency module is useful or not, but there is a lack of visuals to show that the module can capture high-frequency information.
2.	In the ablation experiment, only the DAM was removed, but other simple CNN structures or Transformer structures were not considered to replace the DAM, which did not well illustrate the uniqueness and effectiveness of the module

**Questions:**

1. Can you show some visuals to illustrate that HFCP can capture high-frequency information?
2. Can you replace DAM with other simple structures to illustrate the uniqueness and effectiveness of your DAM?

---

> ### Author Response · Authors · 2024-11-20
> **Response to Reviewer ZvDc**
>
> Thank you for your valuable comments and constructive feedback. We appreciate your efforts in reviewing our work. Below are our responses to your questions:
>
> **1. Regarding Weakness 1 and Question 1:**
>
> We have provided visualizations in the newly submitted manuscript under Appendix B.5 (Figures 9 and 10), where we demonstrate the ability of HFCP to capture high-frequency information. Additionally, we included experimental results in Table 14 (also shown in the following Table 1) to further validate the impact of removing HFCP.
>
>  |Method|Enhancement|Training Strategy|AP(b)|AP(m)|
>  |-|-|-|-|-|
>  |ViT-UWA (Full Model)|-|End-to-End|**44.9**|**42.0**|
>  |ViT-UWA (w/o HFCP)|-|End-to-End|42.9|40.0|
>  |ViT-UWA (w/o HFCP)|FUnIE| Enhance-then-Train|38.1|35.2|
>  |ViT-UWA (w/o HFCP)|NU2Net| Enhance-then-Train|37.3|35.3|
>  |ViT-UWA (w/o HFCP)|PUIE-Net| Enhance-then-Train|34.7|31.6|
>
>  **Table 1:** Different methods of underwater enhancement.
>
> **2. Regarding Weakness 2 and Question 2:**
>
> We have conducted ablation experiments to address this concern and included the results in the newly submitted manuscript under Table 13 of Appendix B.4 (also shown in the following Table 2). We adopt ViT-UWA-T as the basic model. These experiments compare DAM with simple CNN structures and similar-function modules of ViT-Adapter, ViT-CoMer, and InternImage to showcase its superiority.
>
>
>  | Method         | #Param | #FLOPs | AP(b) | AP(m) |
>  |----------------|--------|--------|----------------|----------------|
>  | DAM (ours) | 30M    | 259G   | **40.9**       | **38.8**       |
>  | CNN            | 28M    | 260G   | 38.9           | 37.0           |
>  | SPM            | 29M    | 261G   | 40.3           | 38.1           |
>  | MRFP           | 30M    | 264G   | 39.2           | 37.9           |
>  | Stem + DS      | 31M    | 268G   | 39.6           | 38.2           |
>
>  **Table 2:** Ablation of Detail Aware Module. “DS” means downsampling layers.
>
> As shown in the above Table 2, under a similar scale of parameters, our method achieves the lowest computational cost and the best performance, indicating that DAM can obtain multi-scale features with rich high-frequency details more efficiently.

---

### Official Review · Reviewer_pmDp · 2024-11-04

**Soundness:** 3
**Presentation:** 3
**Contribution:** 3
**Rating:** 8
**Confidence:** 4

**Summary:**

The paper proposes adapted backbone for the Vision Transformer (ViT) architecture termed as Vision Transformer Underwater Adapter (ViT-UWA) specifically designed for dense prediction tasks in images acquired underwater. The two salient features of the proposed ViT-UWA are (a)  its ability to recover and capture high-frequency information in underwater images via the introduction of a high-frequency components prior (HFCP) and (b) it ability to capture features at multiple scales of detail for dense prediction tasks via a detail aware module (DAM) that is based on a multi-scale detail-focused convolutional feature pyramid.

**Strengths:**

The paper proposes a useful enhancement of the standard ViT architecture. The experimental results are encouraging when compared to other state-of-the-art (SOTA) methods for dense prediction in underwater imagery.

**Weaknesses:**

The high frequency signal attenuation observed in underwater imagery is very dependent on the environmental conditions. The proposed method should have incorporated underwater image signal correction that is based on the physics of light scattering under water such as the Sea-Thru algorithm [1].

[1] D. Akkaynak and T. Treibitz, "Sea-Thru: A Method for Removing Water From Underwater Images," 2019 IEEE/CVF Conference on Computer Vision and Pattern Recognition (CVPR), Long Beach, CA, USA, 2019, pp. 1682-1691, doi: 10.1109/CVPR.2019.00178.

**Questions:**

The authors should clarify how they have addressed or would address the underlying physics of underwater signal correction.

---

> ### Author Response · Authors · 2024-11-20
> **Response to Reviewer pmDp**
>
> Thank you for your recognition of our work, and the valuable advice. In the revised manuscript, we have added a discussion on physics-based light scattering correction methods (including work [1]) in the **introduction** section (highlighted in **blue**). Additionally, we conducted a preliminary experiment to attempt the incorporation with physical model correction. Referring to the method in [1], we employed a physical model to simulate light attenuation and backscatter, thereby performing a degree of signal correction on the input underwater images. As shown in Table 1, the box AP improved, likely due to the physical correction module reducing backscatter and light attenuation, which clarifies boundaries between targets and backgrounds in underwater images. However, the mask AP results showed a slight decrease, which may be due to the reduced consistency of internal features within the objects after physical correction. We will continue to explore improvements for this issue in future work, such as introducing certain constraints to minimize corrections within the internal regions of the objects. Moreover, depth information could be utilized to guide the model in learning varying light attenuation [1], enhancing its robustness in complex underwater scenarios.
>
> The experimental results highlight the potential of physical model corrections, and we plan to continue refining these methods in future work. Thank you again for your valuable suggestions.
>
>  | Method | #Param | #FLOPs | AP(b) | AP(m) |
>  |-|-|-|-|-|
>  | ViT-UWA-T | 30M | 259G | 40.9 | **38.8** |
>  | ViT-UWA-T + PSC | 30M | 270G   | **41.3** | 38.5 |
>
>  **Table 1:** Effect of Physical Correction Module. “PSC” means Physical Signal Correction.
>
> [1] D. Akkaynak and T. Treibitz, "Sea-Thru: A Method for Removing Water From Underwater Images," 2019 IEEE/CVF Conference on Computer Vision and Pattern Recognition (CVPR), Long Beach, CA, USA, 2019, pp. 1682-1691, doi: 10.1109/CVPR.2019.00178.

---

### Author Response · Authors · 2024-11-26
**Global Response**

We sincerely thank all the reviewers for their efforts in reviewing our paper and for providing valuable and insightful feedback.

We are pleased to see that the reviewers found our work to be a **useful enhancement** of the standard ViT architecture, specifically noting its applicability to underwater imagery (`pmDp`, `JkrF`). We also appreciate the recognition of the **extensive experiments** conducted on different datasets, both quantitatively and qualitatively (`pmDp`, `ZvDc`, `nQpN`, `JkrF`), and the **superior visualization results** compared to other methods (`ZvDc`). Furthermore, we are encouraged by the feedback that the paper is **well-written**, the methods are clearly introduced (`nQpN`, `ZvDc`).

In the revised paper and our responses to each reviewer, we have included additional insights, visual results, and ablation experiments to further clarify and strengthen the understanding of our method. We welcome any further questions or comments from the reviewers and are happy to provide additional clarifications or explanations as needed.

---

### Meta-Review · Area_Chair_DTvN · 2024-12-20

**Metareview:**

The paper has received mixed reviews with widely varying scores. Reviewers generally commend the authors for the comprehensive quantitative and qualitative results obtained across a large number of datasets, as well as for the clear and well-organized presentation of the paper.

The primary concerns stem from Reviewer nQpN and JkrF, who argue that the design of HFCP, DAM, and VCIM lacks sufficient novelty. These reviewers suggest that the proposed modules appear to be designed with consideration of existing work, which raises questions about the originality of the approach.

In light of the feedback, while the paper presents valuable insights and demonstrates solid results, the concerns about the novelty and innovation of the proposed methods cannot be overlooked. Thus, based on the current evaluation, the paper does not meet the threshold for acceptance.

**Additional Comments On Reviewer Discussion:**

All reviewers acknowledge the experimental results presented in this manuscript, which are commendable. However, the primary concern lies in the novelty of the approach, with reviewers questioning the overall contribution of the paper. The authors' rebuttal has not fully addressed these concerns.

Reviewer nQpN raised specific issues regarding the motivation behind the proposed modules HFCP, DAM, and VCIM, and questioned why ViT-UWA performs better in underwater scenarios. While the authors' rebuttal provided further clarification on these points, the reviewer still feels that the explanations are not entirely convincing. As a result, Reviewer nQpN increased their score from 3 to 5.

Despite addressing some concerns in the rebuttal, the overall contribution of the paper still seems insufficient in terms of novelty and innovation. Thus, while the experimental results are solid, the paper does not fully meet the criteria for acceptance.

---

### Decision · Program_Chairs · 2025-01-22

Reject